# Successful Correction by Prime Editing of a Mutation in the *RYR1* Gene Responsible for a Myopathy

**DOI:** 10.3390/cells13010031

**Published:** 2023-12-22

**Authors:** Kelly Godbout, Joël Rousseau, Jacques P. Tremblay

**Affiliations:** 1Molecular Biology Department, Laval University, Quebec, QC G1V 0A6, Canada; kelly.godbout@crchudequebec.ulaval.ca; 2CHU de Québec Research Center, Laval University, Quebec, QC G1V 4G2, Canada; joel.rousseau@crchudequebec.ulaval.ca

**Keywords:** prime editing, CRISPR/Cas9, gene editing, gene therapy, RYR1-related diseases, neuromuscular diseases, *RYR1* gene, mutations

## Abstract

We report the first correction from prime editing a mutation in the *RYR1* gene, paving the way to gene therapies for RYR1-related myopathies. The *RYR1* gene codes for a calcium channel named Ryanodine receptor 1, which is expressed in skeletal muscle fibers. The failure of this channel causes muscle weakness in patients, which leads to motor disabilities. Currently, there are no effective treatments for these diseases, which are mainly caused by point mutations. Prime editing allows for the modification of precise nucleotides in the DNA. Our results showed a 59% correction rate of the T4709M mutation in the *RYR1* gene in human myoblasts by RNA delivery of the prime editing components. It is to be noted that T4709M is recessive and, thus, persons having a heterozygous mutation are healthy. These results are the first demonstration that correcting mutations in the RYR1 gene is possible.

## 1. Introduction

Gene therapy holds incredible promise for treating genetic diseases as it directly addresses the root of the problem. Gene editing can potentially cure thousands of hereditary diseases by correcting mutations. The discovery of CRISPR/Cas9 in 2012 marked a breakthrough in the development of gene therapies [1]. In 2019, Anzalone et al. published an outstanding technique called prime editing [2]. This system can perform all possible nucleotide substitutions, targeted insertions and deletions. This technology enables DNA modification with unprecedented precision and offers significant benefits over the traditional CRISPR/Cas9 and base editing systems [3]. Prime editing technology is composed of a Cas9 nickase fused in the C-terminus with a reverse transcriptase (RT) and uses a prime editing guide RNA (pegRNA) (Figure 1). Cas9 nickase fused with the reverse transcriptase forms the prime editor (PE). The pegRNA is a modified sgRNA and consists of (1) the sgRNA spacer sequence, (2) the scaffold, a common region of the sgRNA that binds to the SpCas9 fused with a reverse transcriptase, (3) a primer binding site (PBS) and (4) a reverse-transcriptase template (RTT), which contains the desired edits [2] (Figure 1). To prevent the rapid degradation of the PBS and RTT fragments by exonucleases, an additional 3′ RNA sequence forming double-strand loops has been added [4]. These modified pegRNAs are called engineered pegRNA (epegRNA).

Many strategies have been developed for prime editing, numbered PE1, PE2, PE3, PE4, PE5 and PE6 [5]. For a PE2 gene modification, the complex formed by the Prime editor and the epegRNA initially binds to the genome, guided by the pegRNA spacer sequence. The SpCas9 nickase (H840A) recognizes a PAM and induces a single-strand break 3 nt upstream [2]. Next, the PBS sequence binds to its complementary sequence on the cut strand. The reverse transcriptase then uses the RTT sequence as a template to synthesize a new DNA strand. At this stage, one of the strands has a duplicated section (5′ flap and 3′ flap), so the cell must eliminate one of the two flaps to reconstitute the double-stranded DNA. If a 3′ flap is kept, the correction will be retained, but editing will be lost if a 5′ flap is kept [2].

To edit the DNA, the desired correction must be introduced in the RTT sequence, as it provides the template for transcription. 

PE3 is similar to PE2 described above but uses an additional nicking sgRNA (nsgRNA) to induce a nick in the strand not initially cut by the SpCas9n-RT guided by the pegRNA. This additional cut forces the cell to rely on the DNA strand initially cut by the SpCas9nRT as a template. This increases the probability of retaining the edit at the mismatch repair stage. PE3 has been shown to increase prime editing efficiency compared to the PE2 method [5].

The prime editing technology may be used to correct mutations causing RYR1-related myopathies. This group of diseases includes malignant hyperthermia (MH), central core disease (CCD), multi-mini core disease (MmD), centronuclear myopathy (CNM), congenital fiber-type disproportion (CFTD) and exertional rhabdomyolysis (ERM), to name a few [6]. Those diseases all have in common that they are due to a mutation in the *RYR1* gene. More than 700 variants in the *RYR1* gene have been identified [7]. This gene codes for a protein called “ryanodine receptor 1” (RyR1), which is the principal calcium channel in the sarcoplasmic reticulum (SR) in skeletal muscle fibers [6] (Figure 2). The dysfunction of this protein affects the flow of calcium into muscles. Various mutations at different positions in the gene will differently affect the RyR1 protein, but mutations in this gene mostly lead to a calcium leak. Depending on the location of the mutated site, RYR1-related diseases may be autosomal dominant or recessive. These disorders are associated with a wide spectrum of clinical phenotypes, including cramps, exhaustion, heat stroke, breathing difficulties, exercise-induced myalgias and malignant hyperthermia. Patients can also present proximal muscle weakness, fatigue, low muscle tone and slow contraction, leading to mild, slow- or non-progressive disabilities [8]. These myopathies, therefore, severely affect the quality of life of patients. The RyR1 protein has limited functional variations, and the *RYR1* gene is one of the most intolerant to sequence variations in the human genome [9].

So far, there is no effective treatment for these RYR1-related diseases. Since many mutations in the *RYR1* genes are point mutations, the results described in the present article clearly demonstrate that prime editing can be used to correct them since it can substitute any nucleotide in the genome.

The present article reports the correction of one of these mutations (i.e., the T4709M) as an example. This particular mutation was selected because there is a mouse model (Ryr1TM/Indel) with that mutation that develops clear symptoms [10]. The T4709M mutation in the *RYR1* gene has been reported several times, and it is associated with muscle weakness (predominant proximal weakness and marked facial weakness), scoliosis, ophthalmoplegia and respiratory impairment [11]. It has also been associated with multi-minicore disease [12], minicore myopathy with external ophthalmoplegia [11,13,14], susceptibility to malignant hyperthermia, central core myopathy and congenital multicore myopathy with external ophthalmoplegia [15].

## 2. Materials and Methods

### 2.1. Cell Lines

WT human myoblasts were collected from a cadaveric donor who did not have an RYR1-related disorder. Fibroblasts derived from a skin biopsy of a patient with a heterozygous T4709M mutation were obtained from Dr Dowling’s Laboratory.

### 2.2. Plasmids

The pCMV-PE2 (prime editor) and pU6-tevopreq1-GG-acceptor plasmids were bought from AddGene (AddGene plasmids #132775 and #174038). Cloning in these plasmids was carried out as described by Anzalone et al. [2] to make the epegRNA plasmids. Our laboratory, however, modified the plasmid #174038 to include a second U6 promoter and a cloning site to insert the nsgRNA for the PE3 method (plasmid subsequently called epegRNA-nsgRNA). Oligonucleotides used for constructing epegRNAs and nsgRNAs were purchased from IDT Inc. (Coralville, IO, USA). All spacer, PBS, RTT and nsgRNA sequences are in Appendix A.

### 2.3. Cell Culture

HEK293T cells and human fibroblasts were cultured in DMEM medium (Wisent Inc., Saint-Jean-Baptiste, QC, Canada) supplemented with 10% FBS (Wisent Inc.) and 1% penicillin–streptomycin (Wisent Inc.). Cells were kept at 37 °C with 5% CO_2_ in a humidified incubator.

The human myoblasts were cultured in a homemade medium made of DMEM medium with 20% FBS (Wisent Inc.), 16% of medium 199 (Invitrogen™ Inc., Carlsbad, CA, USA) and 1% penicillin–streptomycin (Wisent Inc.), supplemented with Fetuin 25 μg/mL (Life Technologies, Carlsbad, CA, USA), hEGF: 5 ng/mL (Life Technologies), bFGF: 0.5 ng/mL (Life Technologies), insulin: 5 μg/mL (Sigma-Aldrich Canada Inc., Oakville, ON, Canada, 91077C-1G) and Dex: 0.2 μg/mL (Sigma-Aldrich Canada Inc.).

### 2.4. Transfection of HEK293T Cells

The day before transfection, HEK293T cells were detached from the flask with a Trypsin–EDTA solution (Sigma-Aldrich Canada Inc., Oakville, ON, Canada), counted and plated in a 24-well plate at a density of 60,000 cells per well with 500 μL of culture medium. On the day of transfection, 1 μg of total DNA (500 ng of the plasmid coding for prime editor and 500 nf of the plasmid coding for the epegRNA and the nsgRNA) was transfected with Lipofectamine 2000 (Invitrogen™ Inc., Carlsbad, CA, USA) according to the manufacturer’s instruction. Control transfection with eGFP plasmid has also been performed. The medium was changed to 1 mL of fresh medium 24 h later, and cells were maintained in incubation for 48 h before genomic DNA extraction.

### 2.5. Plasmid Electroporation in Myoblasts and Fibroblasts

A total of 2 μg of plasmids (1 μg of prime editor plasmid and 1 μg of epegRNA-nsgRNA plasmid (containing the epegRNA sequence and the nsgRNA for PE3)) was added to 100,000 human myoblasts or fibroblasts and electroporated with the 10 μL Neon Transfection System using the following program: 1100 volts; 20 ms; 2 pulses. Control electroporation with eGFP plasmid was also performed. These electroporated cells were added to 1 well of a 24-well culture plate containing 500 μL of the homemade medium. The medium was changed with 1 mL of fresh medium after 24 h. Cells were maintained in incubation for 48 h before genomic DNA extraction.

### 2.6. PE mRNA In Vitro Transcription

The prime editor plasmid was first amplified by PCR to add optimized 5′ UTR regions and a poly-A tail in 3′ (Appendix A). The resulting PCR product was purified with the PCR Products Purification Kit (EZ-10 Spin Column) (Bio Basic, Toronto, ON, Canada) and served as a template for subsequent in vitro transcription. Prime editor mRNAs were transcribed from this template using the HiScribe T7 mRNA Kit with CleanCap Reagent AG (New England BioLabs Inc., Ipswich, MA, USA) with complete replacement of UTP with N1-Methylpseudouridine-triphosphate (TriLink Biotechnologies Inc., San Diego, CA, USA). The reaction mixture was incubated at 37 °C for 4 h. The reaction volume was then increased to 50 µL with nuclease-free water. Then, 2 µL of DNase I was added to the reaction mixture. The reaction mixture was incubated at 37 °C for 15 min. Transcribed mRNAs were purified with the Monarch RNA Cleanup Kit (500 µg) (New England BioLabs Inc., Ipswich, MA, USA), eluted in 1 mM Sodium Citrate pH 6.4 and quantified by BioDrop, Cambridge, UK. PE mRNAs were then stored at −80 °C.

### 2.7. RNA Electroporation in Myoblasts

A total of 3.7 μg of RNA (0.5 μg of PE mRNA, 2.3 μg of epegRNA (chemically synthetized by IDT Inc., Coralville, IA, USA)) and 0.9 μg of nsgRNA (chemically synthetized by IDT Inc., Coralville, IA, USA) were added to 50,000 human myoblasts and electroporated with the 10 μL Neon Transfection System (program: 1100 volts/20 ms/2 pulses). These electroporated cells were added to 1 well of a 24-well culture plate containing 500 μL of the homemade medium. Control electroporation with eGFP mRNA (synthesized by IVT) was also performed. The electroporation medium was changed with 1 mL of fresh medium after 24 h. Cells were detached with trypsin and harvested in 1 mL culture medium 48 h later, and DNA was extracted from these cells. For multiple treatments, three replicates were placed in 1 well of a 6-well culture plate containing 3 mL of the homemade medium after electroporation. Next, 1 mL of this well was taken and put in 1 well of a 24-well plate. One additional mL of homemade medium was added to wells in the 6-well plate to reach a total of 3 mL of medium per well. The medium was changed after 24 h. Cells in the 24-well plate were harvested for DNA extraction after 48 h. When cells in the 6-well plate were confluent, they were passed in a T25 flask. When confluent, a sample of these cells was treated again via electroporation.

### 2.8. Genomic DNA Preparation and PCR Amplification

HEK293T cells were detached from wells directly with up and down pipetting of the culture medium and transferred in 1.5 mL Eppendorf tubes. Cells were centrifuged for 5 minutes at 8000 RPM at room temperature. Cell pellets were washed with 500 μL of PBS and centrifuged again for 5 min at 8000 RPM. A mix containing 50 μL of DirectPCR Lysis Reagent (Viagen Biotech Inc., Los Angeles, CA, USA) and 0.5 μL of a proteinase K solution (20 mg/mL) for each sample was prepared. Then, 50.5 μL of this mixture was added to each cell pellet. Myoblast cells were washed with 500 μL of PBS in their well. A mix of 100 μL of DirectPCR Lysis Reagent and 1 μL of a proteinase K solution (20 mg/mL) was added to each well. For HEK293T cells and myoblasts, samples were incubated for 2 h at 56 °C followed by 45 min at 85 °C. Samples were then centrifugated at 13,000 RPM for 5 min. Next, 1 or 2 μL of each genomic DNA preparation (supernatant) was used for the PCR reaction. PCR temperature cycling was performed as follows: 30s at 98 °C, followed by 35 cycles of (10 s at 98 °C, 20 s at 60 °C and 30 s at 72 °C) and concluded with 5 min at 72 °C. Phusion™ High-Fidelity DNA polymerase from Thermo Scientific Inc. (Waltham, MA, USA) was used for all PCR reactions. The primers used for the PCR are listed in Appendix A.

### 2.9. Sanger Sequencing

Amplicons from PCR were sent for Sanger sequencing to the sequencing platform of the CHU de Québec Research Center (https://sequences.ulaval.ca/murin/servseq.pageaccueil, accessed on 16 May 2022 to 16 November 2023). Polymerization using the BigDye™ Terminator v3.1, Thermo Fisher Scientific, Waltham, MA, USA was made with an internal primer (Appendix A). Sequences were analyzed with the EditR online program (https://moriaritylab.shinyapps.io/editr_v10/, accessed on 18 May 2022 to 18 November 2023) [16] to analyze editing efficiencies.

### 2.10. Deep Sequencing Analysis

Deep sequencing samples were prepared following the PCR protocol mentioned above but with different primers (Appendix A) for subsequent deep sequencing. PCR samples of 300 bp amplicon length were sequenced with the Illumina sequencer at the Genome Quebec Innovation Centre at McGill University (https://cesgq.com/en-services#en-sequencing, accessed on 19 October 2022). The CRISPResso 2 online program (http://crispresso2.pinellolab.org/submission, accessed on 17 April 2023) was used according to the authors’ guidelines [17] to analyze the results.

### 2.11. Statistical Analysis

Data were analyzed using the GraphPad PRISM 10.0.3 software package (Graph Pad Software Inc., La Jolla, CA, USA). 

## 3. Results

As an initial step in designing a prime editing system targeting codon 4709, we set out to compare the efficiency with which the PE2 and PE3 systems (with the original PE editor or PEmax) could introduce a silent mutation at this site. We used epegRNA containing the tevopreQ1 motif at the 3′ RNA sequence to prevent the rapid degradation of the PBS and RTT fragments by exonucleases [4].

### 3.1. Optimization of Prime Editing: Comparing the PE3 vs. PE2 Strategy’s Effectiveness

The PE3 strategy significantly increased the percentage of editing (Figure 3a,b). This strategy increases editing by about two folds (when using pegRNA with a 10 or 13 nt PBS length) and by close to four folds (when using pegRNA with a 16 nt PBS). Therefore, for the following experiments, the PE3 strategy was employed. 

PEmax is a modified version of the PE protein that was constructed with the aim of increasing the efficiency of prime editing [5]. Although Chen et al. [5] demonstrated that PEmax had a superior efficacy over the original PE, PEmax did not give superior efficacy in our case (Figure 3c,d). However, it was interesting to note that the efficacy ratios of PE3 to PE2 gave similar values, regardless of whether PEmax or original PE was used.

### 3.2. Optimizing Prime Editing: Effect of Varying PBS and RTT Lengths on Editing Efficiency

Several versions of the epegRNA plasmid were constructed to test different PBS and RTT lengths. Although these epegRNAs were tested under different conditions (PE2, PE3, PEmax) and in different cell types (HEK293T and myoblasts), the emerging pattern was very similar (Figure 4a). The RTT length impacted the efficiency of the prime editing. In our case, using an RTT between 14 and 22 nt gave the best results (Figure 4b). The length of the PBS sequence also significantly impacted the percentage of editing obtained. A shorter PBS, i.e., 10 nt, gave the best editing rates. The subsequent experiments were, thus, conducted with six different epegRNAs (Table 1). Moreover, as the mutation to be corrected could not be followed in cells that did not contain the mutation (such as HEK293T cells and normal myoblasts), we also inserted a silent mutation in a nucleotide adjacent to the eventual patient mutation to be studied (Figure 5). This enabled us to assess the efficacy of our epegRNAs, even in WT cells.

### 3.3. Correction of the T4709M Mutation in Patient Fibroblasts

Fibroblasts derived from a skin biopsy of a patient with a heterozygous T4709M mutation were obtained from Dr Dowling’s Laboratory. Surprisingly, no significant editing was observed in these cells, even by Illumina deep sequencing (Figure 6).

### 3.4. Insertion of Silent Mutation in the T4709M Codon in the RYR1 Gene in Normal Myoblasts

The prime editing treatment was subsequently tested in normal myoblasts. The gene editing was monitored by inserting a silent mutation of a nucleotide adjacent to the eventual patient’s mutation.

Adding a silent mutation to the PAM sequence is a strategy that increases the editing percentage by preventing the SpCas9n-RT from binding to an already edited *RYR1* gene. However, the mutated sequences of the *RYR1* gene currently under investigation in our laboratory did not allow us to insert a silent mutation in the PAM sequence. The PAM was located at the junction between the end of exon 96 and the beginning of the next intron. Inducing a substitution in this region would, therefore, result in a modification of the splicing site, which could have a detrimental effect on intact mRNA expression.

Mbakam et al. [18] demonstrated that the introduction of additional silent mutations in the RTT sequence increased editing efficiency, as it allowed the bypass of the mismatch repair system. Liu’s group also reached the same conclusion [5]. Therefore, new epegRNAs 3C, 3D, 3E and 3G were constructed to test this strategy for the *RYR1* gene (Figure 7 and Table 2). We also tested the hypothesis that mutating the first two nucleotides (AA) of the RTT (which are also common to the spacer) (Figure 7) would simulate the effect of a mutation in the PAM. Our hypothesis is, therefore, that these mutations would limit subsequent spacer attachment, an effect that may be mainly observable after multiple treatments. Introducing mutations into the RTT sequence that are also common to the spacer sequence (the first three nucleotides of the RTT) may also increase prime editing efficiency as it could reduce the misfolded epegRNA interactions [19].

Further, 23% editing (i.e., insertion of the silent mutation) was obtained following a single treatment with epegRNA3. All epegRNAs with a PBS of 10 nucleotides (epeg2, 3, 3C, 3E and 3G) were significantly more effective than epegRNAs with a PBS of 13 nucleotides (epeg 4, 5 and 6) (Figure 8).

### 3.5. Generation of a Cell Line Homozygous for the T4709M Mutation in the RYR1 Gene by Prime Editing of WT Myoblasts

Since it was important to verify whether it was possible to edit the patient’s mutation in human myoblasts, and since we did not have access to a patient’s muscle biopsy, a myoblast cell line containing a homozygous *RYR1* T4709M mutation was created by prime editing on normal myoblasts (Figure 9).

### 3.6. Correction by Prime Editing of the T4709M Mutation in the RYR1 Gene in a Myoblast Cell Line Possessing a Homozygous Mutation 

The prime editing treatment was then tested in this homozygous cell line (Figure 10a). After a single treatment, epegRNAs 2, 3, 3C, 3D and 3E corrected, on average, between 10 and 13% of the T4709M mutation in the *RYR1* gene. The efficacy of these epegRNAs was not significantly different from each other but was significantly different from the negative control.

### 3.7. Correction of the T4709M Mutation by RNA Delivery

Delivering prime editing components as RNAs resulted in higher correction rates than delivering plasmid DNA. Indeed, 38% of correction of the T4709M mutation was reached in the *RYR1* gene in the myoblast cell line possessing this mutation in the homozygous state (Figure 10c). The PE mRNAs were produced by in vitro transcription, and three different 5′ Caps were tested: the ARCA Cap [20], the CleanCapAG [21] and the new CleanCapM6 [22]. The two first caps were initially compared in HEK293T cells, and the CleanCapAG was the best of them (Figure 10b). We also tried another version of the CleanCap, the CleanCapM6, which is a new cap released by Trilinks in 2023. In our case, for prime editing, this new CleanCapM6 was not efficient (RNA V3 in Figure 10c). Therefore, the CleanCapAG was used for the subsequent prime editing experiments using PE mRNAs. For experiments by RNA delivery, the epegRNA used was version 3, which had a PBS of 10 nt and an RTT of 16 nt and contained the correction of the T4709M mutation and a silent edit in the codon 4709.

Three successive treatments of prime editing were tried to increase the correction efficiency. In this experiment, 29% of corrections were obtained after the first treatment (Figure 10d). The editing rate increased to 42% after the second treatment (Figure 10d). Up to 59% of corrections were reached after the third treatment, with an average correction rate of 55%.

## 4. Discussion

As prime editing can perform insertions, deletions and all types of substitutions, it can potentially correct all point mutations that cause inherited diseases. Prime editing can also generate cell lines or animal models for specific mutations, thereby driving forward research into many genetic diseases [3]. To date, many in vitro and in vivo preclinical studies have shown that prime editing enables the precise correction of mutations in the genome [3]. In the field of human gene therapy, prime editing has been attempted in vitro and in vivo for hereditary liver diseases (DGAT1 deficiency [23], bile salt export pump deficiency [23], alpha-1-antitrypsin deficiency [24,25], phenylketonuria [26] and tyrosinemia type 1 [27,28,29]), eye diseases (Leber congenital amaurosis [27], Cataracts [30] and X-linked retinitis pigmentosa [31]), skin diseases (recessive dystrophic epidermolysis bullosa [32] and Oculocutaneous albinism [33]), Duchenne muscular dystrophy [18,34] and neurodegenerative diseases (Tay–Sachs disease [35] and Alzheimer’s Disease [36]), as well as cystic fibrosis [37], beta-thalassemia [38], sickle cell disease [39], X-linked severe combined immunodeficiency [40] and cancer [33,41].

The present work is the first demonstration of an efficient preclinical gene therapy for RYR1-related diseases in the scientific literature.

As indicated in the original article of Anzalone et al. [2], for each point mutation to be corrected, the best epegRNA sequence has to be identified by trying several epegRNAs with different lengths of PBS and RTT. In this study, the length of the PBS was more restrictive than the length of the RTT. Varying the PBS length from 10 nt, 13 nt and 16 nt heavily impacts efficiency. Thus, for the T4709M locus, a smaller PBS of 10 nt was the best option. A recent article recommended a PBS of 7 nt to improve prime editing by preventing inhibitory interactions between the PBS and the spacer sequences of the epegRNA [42]. For the RTT, many lengths gave the same efficiency. RTT of 14 nt, 16 nt, 19 nt and 22 nt gave the best efficiency and were not significantly different. As prime editing optimization seems to be sequence-specific, those conclusions may not apply to other targets [16,43]. 

No significant editing of the T4709M mutation of the *RYR1* gene was obtained in the patient fibroblasts. Our hypothesis is that since the RYR1 gene is not expressed in fibroblasts [44] but only in skeletal muscles, the region to edit must not be easily accessible to prime editing components. However, this experimental constraint will be an advantage in future clinical applications. In fact, genome editing will only be carried out in skeletal muscles, so the off-target risks in other tissues are practically offset.

Testing the prime editing treatment in WT myoblasts allowed us to narrow the epegRNA choice for the next steps. EpegRNAs with a PBS of 10 nt outperformed epegRNAs with a PBS of 13 nt. Thus, for the next experiments, epegRNAs with a PBS of 10 nt will be used since 23% of editing was obtained in WT myoblasts and more than 40% in HEK293T cells. This result is not surprising as HEK293T cells are easy to transfect and have a deficient mismatch repair system [44], which increases prime editing efficiency.

A strategy to bypass the mismatch repair system of the cells is to introduce additional silent mutations in the RTT [5,18]. Therefore, new epegRNAs 3C, 3D, 3E and 3G were constructed to test this strategy for the *RYR1* gene (Figure 7 and Table 2). However, adding additional silent mutations did not lead to an improvement. We also tested the hypothesis that mutating the first two nucleotides (AA) of the RTT (which are also common to the spacer) (Figure 8) would simulate the effect of a mutation in the PAM. Therefore, we hypothesized that these mutations would limit subsequent spacer attachment. However, editing rates were not significantly different from the original constructions that did not include additional edits.

While generating the myoblast cell line with the T4709M mutation in the *RYR1* gene, the manual cloning allowed us to detect an unwanted rare event. In addition to the desired mutation, we observed an undesired substitution at the heterozygous state. This indel was created because the reverse transcriptase continued to reverse transcribe, even after the end of the RTT sequence. In the epegRNA, the RTT sequence is followed by the epegRNA scaffold sequence that binds to the Cas9 protein. The first nucleotide of this scaffold sequence was incorporated into the T4709M mutated cell line. Chen et al. [5] also observed this kind of off-target. The scaffold sequence should be optimized to avoid this type of off-target event. The sequence could be modified as the first part of the scaffold sequence after the RTT makes a stronger double-stranded RNA by inserting C and G nucleotides at adequate positions. Providing a longer RTT sequence may also be investigated. Genome-wide research for off-target editing was not conducted here for the in vitro experiments. Still, it would be an important step to validate eventually, especially in in vivo preclinical studies.

In the myoblast cell line possessing the T4709M mutation in the *RYR1* gene in the homozygous state, 10 to 13% of corrections were obtained. The adjacent silent mutation had the same editing rate. The lower prime editing efficiency in that cell line compared to in WT myoblasts may be explained by the history of the cells. The T4709M cell line was generated by four electroporation treatments followed by manual cloning. Those events affected the cells and may have made them more sensitive to additional electroporation treatments. 

The form in which the prime editing components are delivered can significantly impact editing efficiency. For example, Li et al. [45] used prime editing to introduce substitutions in the SNCA gene in human pluripotent stem cells (hPSCs). They compared the efficiency when using a PE2 treatment by delivering the components as the plasmids, mRNAs or ribonucleoproteins (RNPs) and obtained about 5%, 26.7% and 1% editing efficiency, respectively. Our results point to the same conclusion. Delivering the prime editing components as RNAs dramatically increased the prime editing efficiency and cell viability as RNA is less toxic to cells than DNA.

Up to 59% of corrections were obtained by RNA delivery in the T4709M myoblast cell line. This result is encouraging since persons heterozygous for this mutation are healthy.

## 5. Conclusions

Our results are the first demonstration that correcting mutations in the RYR1 gene is possible. This project paves the way for treating RYR1-related myopathies using gene therapy. Achieving a correction rate of more than 50% in myoblasts is, therefore, possible, opening the way to correct recessive mutations in the *RYR1* gene. Correcting dominant mutations can still be a target, but it will probably require higher editing efficiency. More prime editing improvements would be needed. Prime editing is also highly sequence-dependent, so the optimization and achievable efficiency can vary depending on the mutation’s position in the gene.

These breakthroughs will significantly impact the treatment of other hereditary diseases. Indeed, this method can be applied to other genetic mutations, and in each case, only the epegRNA would have to be modified.

## Figures and Tables

**Figure 1 cells-13-00031-f001:**
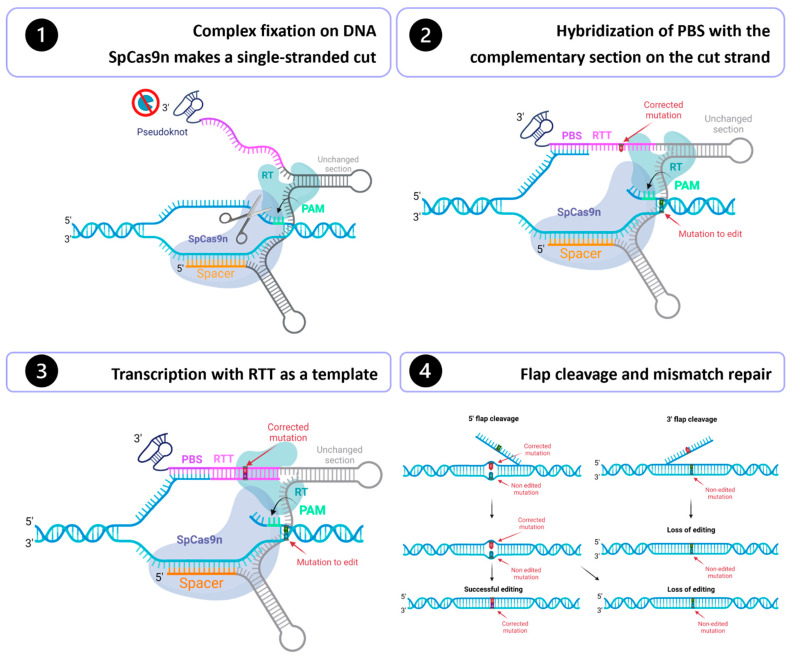
Prime editing mechanism. 1—The spacer sequence of the epegRNA is complementary to a sequence in the genome. This complementarity guides the Prime editor to the right place in the DNA. The Cas9n recognizes a PAM and cuts a single strand 3 nt upstream. 2—The PBS hybridizes to its complementary sequence on the cut strand. 3—The RTT will synthesize the new DNA strand using the RTT as a template. 4—One DNA flap is removed, and the mismatches are repaired. Figure from Godbout et al. [3].

**Figure 2 cells-13-00031-f002:**
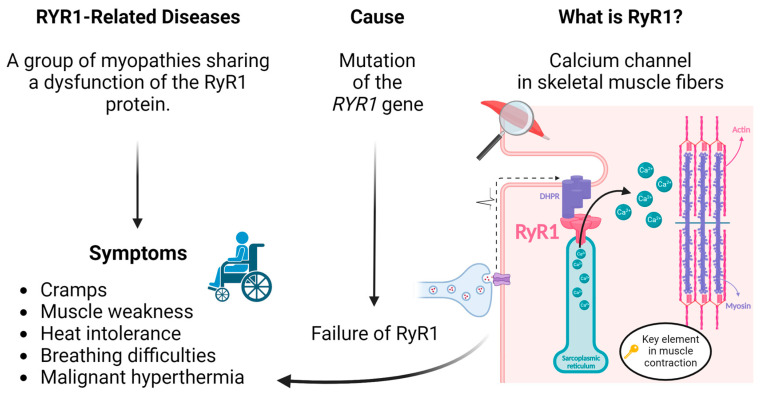
RYR1-related diseases: cause and symptoms.

**Figure 3 cells-13-00031-f003:**
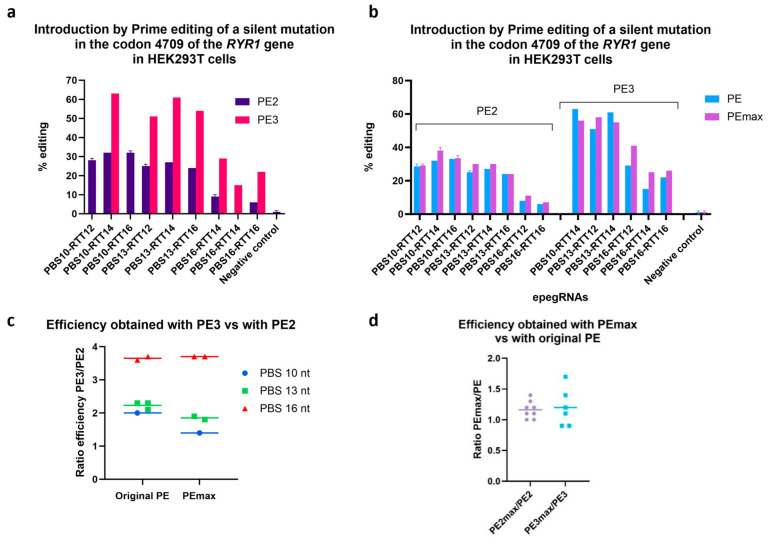
Comparison of the efficiency obtained using the PE2, PE3 and PEmax strategy. The efficiency of various technological variations of the Prime editing technique was tested by inserting a silent mutation in codon 4709 of the *RYR1* gene in HEK293T cells. (**a**) Illustration of the results using the PE2 and the PE3 methods. (**b**) Illustration of the PE2 and PE3 methods using the PE and the PEmax strategies. (**c**) The figure illustrates the ratio of the percentage of gene editing obtained with PE3 relative to the percentage obtained with PE2. (**d**) Illustration of the ratio of PE2max editing relative to PE2, and of PE3max relative to PE3.

**Figure 4 cells-13-00031-f004:**
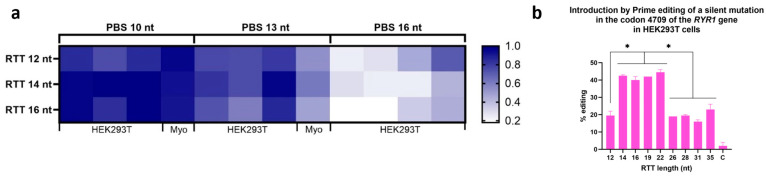
Effect of varying PBS length and RTT on editing efficiency. (**a**) The darker the box is, the higher is the percentage of editing. This heatmap also summarizes the effects of varying the length of RTT and PBS tested under different conditions (PE2, PE3, PE2max, PE3max) and in different cell types (HEK293T and myoblasts), in order to compare only the effects of RTT length and PBS, i.e., without considering the effect of the strategy used or the cell type used. For each sample, the ratio of its efficiency value to the highest efficiency value obtained in the same experiment under the same condition was calculated. A value of 1 thus indicates that this sample obtained the best editing efficiency in the experiment. In order from left to right, the columns represent the following conditions for 10 nt PBS and 13 nt PBS: HEK293T PE2, HEK293T PE3, HEK293T PE2max and Myoblasts PE3. For 16 nt PBS, the columns are as follows: HEK293T PE2, HEK293T PE3, HEK293T PE2max and HEK293T PE3max. Since epegRNAs with 16 nt PBS were significantly less effective in experiments in HEK293T, these epegRNAs were not tested in myoblasts. (**b**) Illustration of the results of inserting a silent mutation in codon 4709 of the *RYR1* gene in HEK293T cells while testing different RTT lengths. All the epegs used in this experiment had a PBS containing 10 nt. The length of the RTT was varied from 12 to 35 nucleotides. RTT containing 14 to 22 nucleotides produced the best Prime editing results. One-way ANOVA and multiple comparisons were used. * represents a significant difference using a *p*-value of 0.001.

**Figure 5 cells-13-00031-f005:**
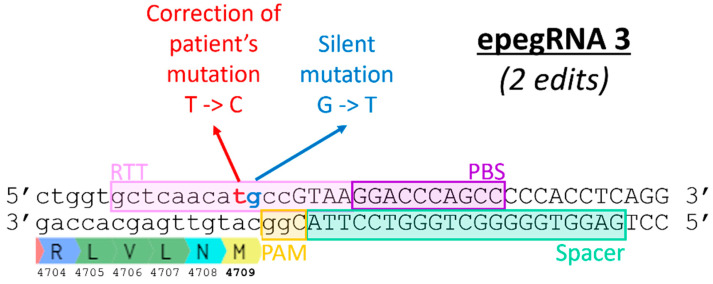
Sequence of the section of the *RYR1* gene containing the T4709M mutation. The coding nucleotides are in minuscules, and those in the intron are in capital letters. The sequences necessary to construct an epegRNA that would correct the T4709M mutation are identified (Spacer, PBS and RTT). Note that the sense strand is on the top of the figure. Thus, the PAM (5′Cgg3′) is in the antisense strand, and it is the antisense strand that will be nicked by the SpCas9 nickase part of the PE. The normal 4709 codon is A**C**G coding for threonine. In the patient, this codon is mutated to A**T**G coding for methionine. For correcting the patient mutation, a **C** will have to be inserted in the RTT sequence (5′ gctcaaca**C**gccGTAA3′) so that the reverse transcriptase synthesizes a new anti-sense strand containing a 5′c**g**t3′ threonine anti-sense codon. However, since the experiment was carried out in HEK293T cells containing the normal *RYR1* gene, no modification could be detected by sequencing when using such an RTT. An additional silent mutation was thus inserted in the RTT (5′ gctcaaca**C**TccGTAA3′) to modify the normal ACG codon into an ACT codon also coding for threonine.

**Figure 6 cells-13-00031-f006:**
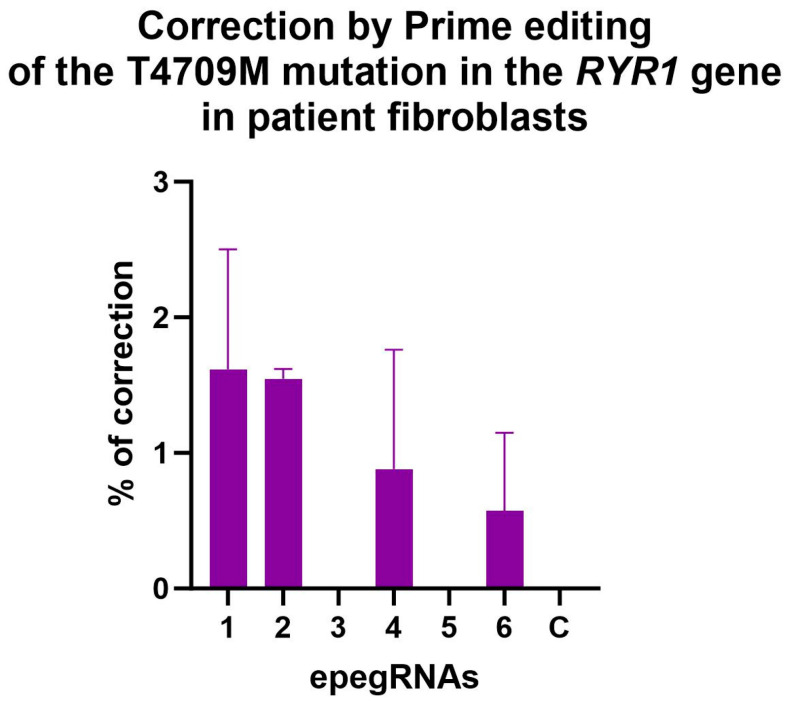
Correction of the T4709M mutation in patient fibroblasts. The percentage of editing was determined by Illumina deep sequencing. No editing was observed with epegRNAs 3 and 5. The editing percentages obtained with the other epegRNAs were not significantly different from the negative control (named C). One-way ANOVA, multiple comparisons and a *p*-value of 0.05 were used.

**Figure 7 cells-13-00031-f007:**
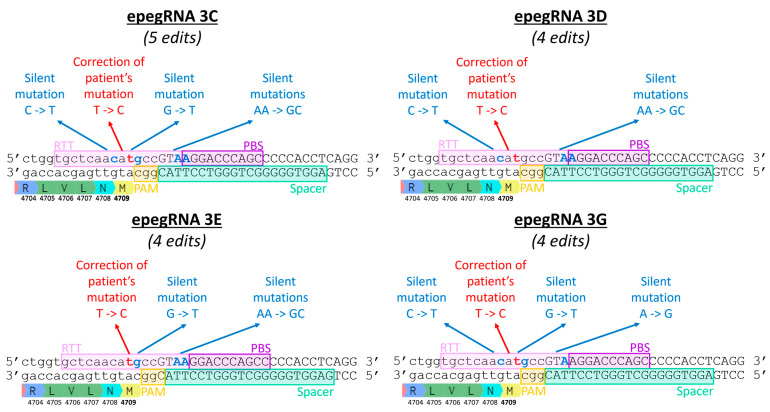
Introduction of additional silent mutations by epegRNAs 3C, 3D, 3E and 3G. The DNA sequence shown is the section of the *RYR1* gene where the T4709M mutation is located. The epegRNA sections (Spacer, PBS, and RTT) are identified. The T4709M mutation is a T instead of a C. Therefore, a C is inserted at this position in the RTT to correct the mutation. Silent mutations are added to epegRNAs in an attempt to increase the efficiency of prime editing.

**Figure 8 cells-13-00031-f008:**
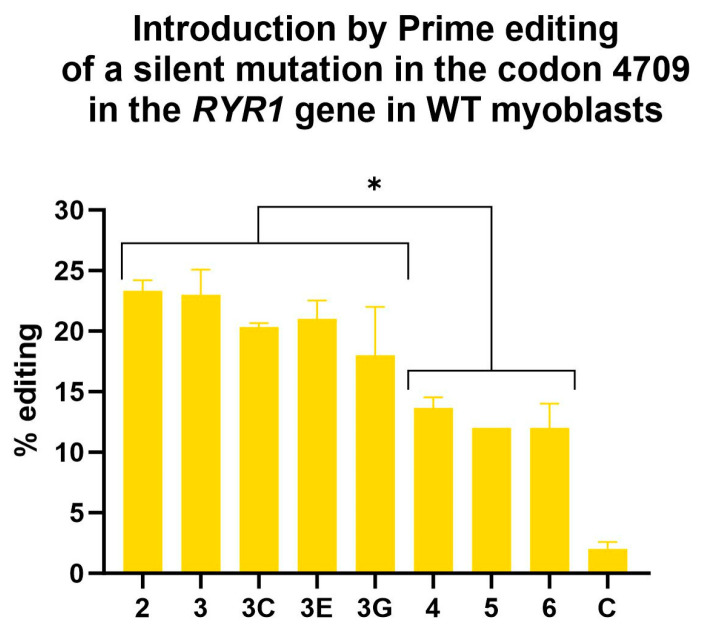
Insertion by Prime editing in wild-type myoblasts of a silent mutation adjacent to the eventual patient’s mutation in the codon 4709 of the *RYR1* gene. A one-way ANOVA indicated that the editing values obtained with epegRNAs 2, 3, 3C, 3E and 3G were significantly different from the editing values obtained with epegRNAs 4, 5 and 6. This test also demonstrated that all epegRNAs produced editing values significantly different from the negative control (named C). An ANOVA test was performed, and * represents a significant difference with a *p*-value of 0.05. The 3D epegRNA is not shown in this graph, as it did not insert the same silent mutation adjacent to the patient’s mutation.

**Figure 9 cells-13-00031-f009:**
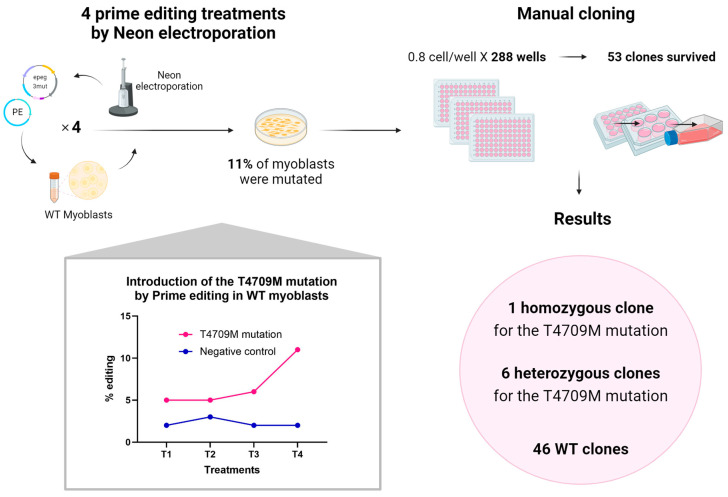
Generation of a cell line homozygous for the RYR1 T4709M mutation by prime editing on WT myoblasts. Four Neon electroporation treatments were performed on immortalized normal myoblasts to deliver the PE components (plasmids coding for PE and an epegRNA). The PE2 strategy was used here to minimize off-targets in the eventual cell line. After these four treatments, the percentage of introduction of the patient’s mutation (T4709M) reached about 11%. Manual cloning was then performed. 0.8 cells were deposited per well in 288 wells of 96-well plates. Only 53 clones survived and multiplied. Of these 53 clones, one turned out to be homozygous for the T4709M mutation, six clones turned out to be heterozygous for this mutation, and 46 clones were WT.

**Figure 10 cells-13-00031-f010:**
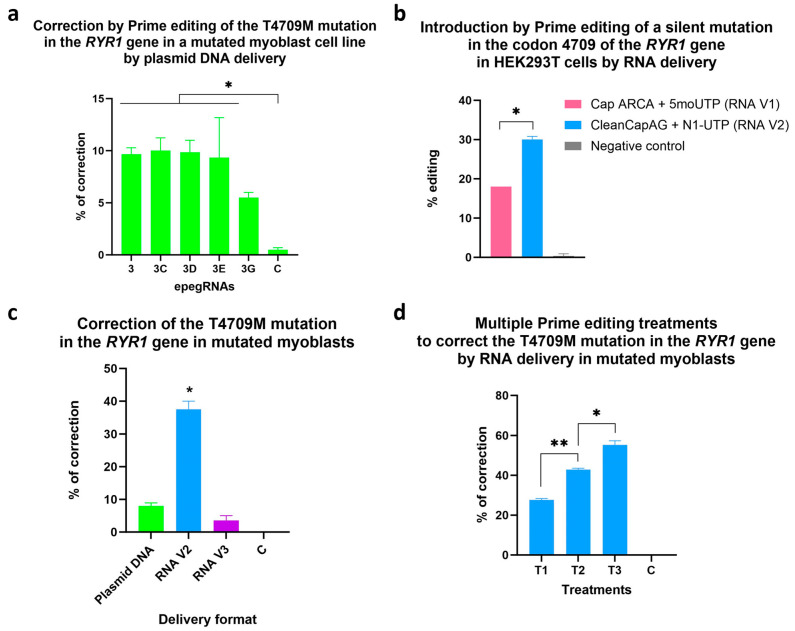
(**a**) Correction by prime editing of the T4709M mutation in the *RYR1* gene in a myoblast line possessing this mutation at the homozygous state by delivering the prime editing components by plasmid DNA. Experiments were performed in triplicates. The figure illustrates the percentage of correction obtained after one treatment of prime editing delivered by plasmid DNA. One-way ANOVA was used as a statistical test. A *p*-value of 0.05 was used. (**b**) Comparison of different formulations for PE mRNA. PE mRNAs were made by in vitro transcription. The first combination (RNA V1) tested was using 5′ Cap ARCA and 5-Methoxy-UTP. The second formulation (RNA V2) was made with the 5′ CleanCapAG and N1-Methylpseudo-UTP. Those PE mRNAs were tested in combination with a chemically synthesized epegRNA in HEK293T cells. The editing rate with the PE3 strategy was evaluated by tracking the silent mutation in the codon 4709 in the *RYR1* gene. (**c**) Correction of the T4709M mutation in the *RYR1* gene by RNA delivery in the mutated myoblast cell line. The percentages of correction were obtained by Sanger sequencing after one Prime editing (PE3) treatment using different delivery methods. Plasmid DNA: PE, epegRNA and nsgRNA were delivered by plasmid DNA. RNA V2: PE mRNA with CleanCap AG and epegRNA and nsgRNA made by IDT. RNA V3: PE mRNA with CleanCapM6 and epegRNA and nsgRNA made by IDT. One-way ANOVA was used as a statistical test. A *p*-value of 0.001 was used. The percentage of correction of RNA V2 was statistically different from all other conditions. (**d**) Multiple prime editing treatments to correct the T4709M mutation in the *RYR1* gene by RNA delivery in the mutated myoblast cell line. The percentages of correction were obtained by Sanger sequencing after one, two and three treatments of Prime editing using the PE3 strategy by RNA delivery. PE mRNA was made with CleanCap AG (RNA V2), and epegRNA and nsgRNA were chemically synthesized by IDT. One-way ANOVA was used as a statistical test. * represents a *p*-value < 0.001 and ** represents a *p*-value < 0.0001.

**Table 1 cells-13-00031-t001:** Sequences and characteristics of epegRNAs 1 to 6. The nucleotide C in red is to correct the patient mutation T4709M, and the nucleotide T in green is to insert a traceable silent mutation.

epegRNAs	PBS Length (nt)	RTT Length (nt)	PBS Sequence	RTT Sequence
1	10	12	GGACCCAGCC	AACACTCCGTAA
2	10	14	GGACCCAGCC	TCAACACTCCGTAA
3	10	16	GGACCCAGCC	GCTCAACACTCCGTAA
4	13	12	GGACCCAGCCCCC	AACACTCCGTAA
5	13	14	GGACCCAGCCCCC	TCAACACTCCGTAA
6	13	16	GGACCCAGCCCCC	GCTCAACACTCCGTAA

**Table 2 cells-13-00031-t002:** Sequences and characteristics of epegRNAs 3, 3C. 3D, 3E and 3G. The nucleotide C in red is to correct the patient mutation T4709M, and the nucleotide T in green is to insert a traceable silent mutation. Additional silent mutations in blue are included in epegRNAs 3C, 3D, 3E and 3G.

epegRNAs	PBS Length (nt)	RTT Length (nt)	PBS Sequence	RTT Sequence
3	10	16	GGACCCAGCC	GCTCAACACTCCGTAA
3C	10	16	GGACCCAGCC	GCTCAATACTCCGTGC
3D	10	16	GGACCCAGCC	GCTCAATACGCCGTGC
3E	10	16	GGACCCAGCC	GCTCAACACTCCGTGC
3G	10	16	GGACCCAGCC	GCTCAATACTCCGTGA

## Data Availability

The data presented in this study are available on request from the corresponding author.

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
