# Peer review of "Successful Correction by Prime Editing of a Mutation in the RYR1 Gene Responsible for a Myopathy"

_cells, 2023, doi:10.3390/cells13010031_

Round 1

Reviewer 1 Report

Comments and Suggestions for Authors

In this manuscript, the authors describe a series of experiments using prime editing to introduce or correct mutations in the RYR1 gene around codon 4709. In a number of optimisation experiments, they evaluate different prime editing systems, PBS lengths, RTT lengths, and RTT designs, achieving in many cases high editing efficiencies. Although similar experiments have been performed by others, it appears that the optimal PE parameters at this locus are in some cases different than has been described at other loci. I think the present work therefore carries helpful lessons for other scientists in this field. Before the manuscript is published however, it would be helpful to address a few points regarding the experiments themselves and the way the data is presented.

Major comments:

* The introduction is fairly lengthy, with one table and 2 figures, and it includes descriptions of the mechanism of action of Cas9, HDR and NHEJ, base editing, etc. Since these topics have been extensively covered in other papers and reviews, I would suggest shortening the introduction overall and focusing on what is known about optimisation of prime editing systems, which is the central theme of the present work.

* Can the authors present data showing efficient delivery of a payload of any kind (editing system or fluorescent reporter) in the patient-derived fibroblasts? Primary cell lines can sometimes be difficult to transfect, and the poor editing results in these cells could simply be attributable to poor delivery of the editing system.

* Along similar lines, have the authors evaluated the efficiency of prime editing system delivery in myoblasts? If only a minority of cells are successfully transfected, this would create a ceiling effect that could explain the largely similar editing efficiency observed with a number of different epegRNAs.

* Can the authors show restoration of RYR1 function in edited cells (e.g. myoblasts)? I think showing evidence of a functional benefit to the achieved editing is an important aspect of evaluating its viability as a therapeutic strategy, since it speaks to the question of how much editing is necessary for a functional benefit.

* Have the authors searched for off-target editing? If not, this should be discussed as a limitation.

In addition, I have a number of other minor comments:

* I would suggest removing the claims of priority ("first ...") in the title and abstract.

* In a few places, the authors describe PE as being able to correct nearly all mutations in the RYR1 gene. I think these statements need to be softened, since (a) the efficiency of prime editing can be highly sequence-dependent, and (b) many RYR1 mutations are dominant and would probably require fairly high in vivo editing efficiency to achieve a therapeutic benefit. This challenge in correcting dominant RYR1 mutations should be mentioned in the discussion.

* The authors describe obtaining myoblasts from "a muscle biopsy of a healthy cadaveric donor". I don't think a cadaver can be described as healthy, and tissue collected at autopsy is not generally referred to as a biopsy, so I would rephrase this as something like "myoblasts were collected from a cadaveric donor who did not have a RYR1-related disorder."

* Which specific 3' motif was added at the pegRNAs to construct the epegRNAs?

* At the very start of the results section, it would be helpful to include a sentence describing the broad outlines of the first experiment that is presented, e.g. something along the lines of "As an initial step in designing a PE system targeting codon 4709, we set out to compare the efficiency with which the PE2 and PE3 systems (with the original PE editor or PEMax) could introduce a silent mutation at this site."

* Do the authors have a hypothesis regarding why the benefit of PE3 over PE2 is more pronounced when tested with shorter PBS lengths?

* Does the patient fibroblast line also carry a second mutation? If not, does the patient have clinical manifestations associated with the single heterozygous T4709M mutation?

* In figure 6 and in the related paragraph of the results section, the pepgRNAs are designated PBS00-RTT00 instead of 1-6, as defined in table 2. Do these indicate different epegRNAs?

* The structure of section 3.4 should be reviewed. This section begins with a presentation of the efficiency of guides 3C-G, but the design of these guides is only explained afterwards.

* What is the denominator when measuring editing efficiency in the patient cell line, which has a heterozygous mutation? Is it total alleles, or mutant alleles?

* I think it would be preferable to report only the means of technical replicates for any given experiment, and avoid using the phrase "up to X%" to report the best of several technical replicates.

* What should be figure 3 seems to be incorrectly labeled as figure 1.

* "% of edition" along the Y axis of several panels and in the text should be corrected to "% editing"

* A few of the panels have faint grey outlines around some edges that should be removed to clean up the figures

Comments on the Quality of English Language

The quality of the English language is overall good, except for a few minor errors like "% of edition", as mentioned above. The manuscript would however benefit from light editing to correct occasional clumsy phrasing (e.g. "the mutation will affect the impact on the protein") and phrases that have an unscientific tone (e.g. "Unfortunately, it also mostly degenerates into motor disability").

Author Response

In the Word file.

Reviewer 2 Report

Comments and Suggestions for Authors

The RYR1 gene codes for Ryanodine receptor 1, which is a calcium channel crucial for skeletal muscle function, and mutations in it lead to muscle weakness and motor disabilities with no effective treatments. Godbout et al. reported the first successful correction of a severe mutation (T4709M) in the RYR1 gene using prime editing, offering a potential avenue for gene therapies in RYR1-related myopathies. Additionally, they achieved a 59% high correction rate of the T4709M mutation in human myoblasts using prime editing, demonstrating the feasibility of correcting mutations in the RYR1 gene. The prime editing approach to correcting the point mutations in the RYR1-related myopathies has tremendous clinical potential. Although this work is mostly performed in an in vitro human myoblast cell line, it will be of great interest to a broad readership for future in vivo applications, especially for those working in the neuromuscular field. There are a few, mostly minor points that need to be addressed.

1. Introduction part page 4 line 90: The authors should introduce more about RYR1-related diseases and talk about the T4709M in the RYR1 gene. For example,'mutations in RYR1 are associated with a wide range of clinical phenotypes, including autosomal recessive RYR1-related myopathies (including congenital myopathy 1B), as well as autosomal dominant myopathies, exercise-induced myalgias, heat stroke, and malignant hyperthermia’. For the T4709M mutation in the RYR1 gene, it has been reported several times (PubMed: 17483490; PubMed: 17033962; PubMed: 23919265; PubMed: 30611313; PubMed: 32403337; PubMed: 31980526; PubMed: 33087929), and it is associated with xxx.

2. The authors did excellent work in characterizing the successful correction of the T4709M mutation in the RYR1 gene. T4709M mutation in the RYR1 gene is associated with a decreased protein level of RYR1 compared with the control (DOI:10.1093/brain/awm096). It would be great if the authors could also do some functional confirmation to compare these homozygous, heterozygous, and WT myoblasts (including western blot, immunofluorescence, or myoblast cell proliferation and differentiation testing).

Author Response

In the Word file.

Round 2

Reviewer 1 Report

Comments and Suggestions for Authors

The authors have adequately addressed my questions. Their replies indicate that they performed control transfections with GFP, but I could not readily find this mentioned in the revised manuscript. If it is not mentioned in the manuscript, I would suggest adding this information in the methods or results.

Comments on the Quality of English Language

English language quality is largely adequate, though the manuscript would benefit from some additional review to correct minor grammatical mistakes and awkward phrasing.

Author Response

Dear reviewer,

I have added the GFP control transfection in the method section. We have also revised our English grammar and phrasing.

Thank you!